# *RcTRP5* Transcription Factor Mediates the Molecular Mechanism of Lignin Biosynthesis Regulation in *R. chrysanthum* against UV-B Stress

**DOI:** 10.3390/ijms25179205

**Published:** 2024-08-24

**Authors:** Fushuai Gong, Wang Yu, Kun Cao, Hongwei Xu, Xiaofu Zhou

**Affiliations:** Jilin Provincial Key Laboratory of Plant Resource Science and Green Production, Jilin Normal University, Siping 136000, Chinakun2199@163.com (K.C.);

**Keywords:** *RcTRP5*, lignin biosynthesis, *R. chrysanthum*, UV-B stress

## Abstract

UV-B stress destroys the photosynthetic system of *Rhododendron chrysanthum* Pall. (*R. chrysanthum*), as manifested by the decrease of photosynthetic efficiency and membrane fluidity, and also promotes the accumulation of lignin. The MYB (v-myb avian myeloblastosis viral oncogene homolog) family of transcription factors can be involved in the response to UV-B stress through the regulation of lignin biosynthesis. This study indicated that both the donor and recipient sides of the *R. chrysanthum* were significantly damaged based on physiological index measurements made using OJIP curves under UV-B stress. The analysis of bioinformatics data revealed that the *RcTRP5* transcription factor exhibits upregulation of acetylation at the K68 site, directly regulating the biosynthesis of lignin. Additionally, there was upregulation of the K43 site and downregulation of the K83 site of the CAD enzyme, as well as upregulation of the K391 site of the PAL enzyme. Based on these findings, we conjectured that the *RcTRP5* transcription factor facilitates acetylation modification of both enzymes, thereby indirectly influencing the biosynthesis of lignin. This study demonstrated that lignin accumulation can alleviate the damage caused by UV-B stress to *R. chrysanthum*, which provides relevant ideas for improving lignin content in plants, and also provides a reference for the study of the metabolic regulation mechanism of other secondary substances.

## 1. Introduction

UV-B stress increases the amount of UV-absorbing materials, damages DNA and membranes, and decreases photosynthesis [1]. Inhibited by UV-B stress, photosystem II in rice leaves is less active, preventing electron transfer, interfering with the photoreduction process of the photosystem II primary electron acceptor Q_A_, and affecting the transfer of electrons from the reaction centers of photosystem II to Q_A_, Q_B_, and PQ. As a result, photosystem II’s potential activity and the efficiency of the primary light energy conversion declines, which in turn reduces the amount of dry matter synthesis and accumulation as well as the efficiency of photosynthesis [2]. In studies on the effects of UV-B stress on plants, the accumulation of UV-absorbing substances as a result of UV-B stress has been extensively examined. These UV-absorbing substances include flavonoids and several other phenolic substances, such as erucic acid esters and flavonoid–erucic-acid ester complexes, which are produced by plants to protect sensitive tissues involved in UV-B stress [3].

The MYB structural domain, which is structurally conserved in all MYB transcription factors, is a family of proteins. The highly conserved N-terminal DNA-binding domain and the varied C-terminal regulatory region of MYB transcription factors are in charge of the protein’s regulatory function. There are four kinds of plant MYB transcription factors that have been identified: 1R-MYB/MYB-related (one R structure); R2R3-MYB (two R structures); R1R2R3-MYB proteins (three R structures); and 4R-MYB (four R structures) [4]. Research has demonstrated the significance of plant MYB transcription factors in secondary metabolism, physiological regulation, growth and development, and other functions [5]. In transgenic *A. thaliana*, the MYB transcription factor *AgMYB5* in celery increases β-carotene production and improves drought resilience. MYB transcription factors contribute to resistance to abiotic stress as well [6]. Exogenous ABA stimulates the Arabidopsis *AtMYB102* gene, and osmotic stress signals control *AtMYB102* expression [7].

Occurring naturally between cellulose and hemicellulose in the cell wall, lignin is a structurally complex macromolecular phenolic polymer that makes up 30% of the organic carbon content of the biosphere [8]. It is a significant derivative of the phenylpropane biosynthesis pathway. Under abiotic stress conditions, lignin helps plants maintain cellular osmotic equilibrium while preserving membrane integrity by reducing osmosis and transpiration of water from the plant cell wall. In addition to having sluggish development and fewer branches, Arabidopsis *AtPrx72* knockout mutants also have much lower levels of lignin, particularly S-type lignin. They also have significantly lower levels of photosynthetic efficiency [9].

A number of plants have been found to carry the genes for the essential enzymes involved in the successive enzymatic reactions that make up lignin production [10]. The first rate-limiting enzyme in the lignin synthesis pathway is phenylalanine ammonia-lyase (PAL, EC 4.3.1.24), and the activity level of PAL directly affects the lignin production process as a whole [11]. The lignin concentration is dramatically decreased and the lignin monomer S/G ratio is raised in the Arabidopsis *AtPAL1/AtPAL2* double mutant [12]. Cinnamyl alcohol dehydrogenase (CAD, EC 1.1.1.195) is primarily responsible for reducing intermediates to lignin monomers [13]. Peroxidases (POD/PRX) are commonly categorized into three main classes: class I is found inside cells, class II is exclusively encoded by fungi, and class III peroxidases (EC 1.11.17) are the subject of this study because of their redox chemistry and ability to catalyze the reduction of H_2_O_2_ [14].

Protein Lys acetylation (LysAc) is a post-translational modification in which the acetyl group is partially transferred to the ε-amino group of the Lys residue, in contrast to the frequent acetylation of the N-terminus of proteins [15]. The lysine site is the most common location for acetylation, and protein acetylation modification typically occurs at the N-terminus of histone proteins. Acetylation can also facilitate the dissociation of histone proteins from the DNA octamer, which allows transcription factors to more effectively bind to DNA and achieve regulatory effects on target genes [16]. In order to realize the regulation of target genes, it can facilitate the dissociation of histones from DNA octamers and improve transcription factors’ ability to bind to DNA. As research and technology advance, lysine acetylation changes are now being discovered in non-histone proteins. Non-histone acetylation is also observed in a range of plants, such as *Arabidopsis thaliana* [17] and rice [18]. The acetyltransferases TBL34 and TBL3, which are necessary for xylan acetylation in Arabidopsis, occur simultaneously. This causes a severe collapse of the plant’s xylem ducts, altering the secondary wall structure and impeding the plant’s growth and development [19].

The conservation of *R. chrysanthum* germplasm resources will benefit from this study, which is also crucial for understanding how metabolite patterns change and for deciphering the regulatory systems behind plant growth adaptation to challenging environments. Predicting the consequences of increasing UV-B radiation on plant communities and ecosystems as a result of ozone depletion requires an understanding of these mechanisms. Food security can be greatly impacted by increasing crop tolerance to UV-B radiation, particularly in regions where crops are exposed to high UV-B radiation levels. Furthermore, knowing how plants adapt to UV-B radiation might aid in the preservation of plant biodiversity, particularly in environments where UV-B radiation fluctuations can have a significant impact.

## 2. Results

### 2.1. OJIP Curves in R. chrysanthum in Response to UV-B Stress

The O and P phases’ variations in OJIP transient curves are depicted in Figure 1A. The findings demonstrated that the amplitude of the I, P, and I–P phases was dramatically decreased by UV-B stress.

To evaluate the multiphase course of the OJIP curves in the O–J, J–I, and I–P phases, the fluorescence data were normalized and displayed as kinetic curves of relative changing fluorescence over time (Figure 1B). Under UV-B stress, no appreciable alterations in the O–J phase were seen. When compared to the control, a sustained decrease in fluorescence intensity was seen in the J–I and I–P phases under UV-B stress. This suggests that the UV-B stress enhanced the ability of the slow-reducing and fast-reducing PQ pools to be reduced, and that it also inhibited receptor-side electron transfer in the PSII reaction center.

The K point was observed by normalizing the relative fluorescence between the O and J phases and calculating the ratio of variable fluorescence to amplitude at the K point (Figure 1C). The observation of a non-significant alteration in W*_K_* suggests that the donor side of the PSII reaction center experienced less damage from UV-B stress compared to the acceptor side.

### 2.2. Reaction of JIP-Measured Parameters in R. chrysanthum to UV-B Stress

Four typical JIP-measured parameters were chosen to reflect the changes in the activity of the electron transport chain in order to better understand the changes in the photosynthetic system of *R. chrysanthum* under UV-B stress. These characteristics include: Fv/Fo, which is related to the activity of the PSII reaction center’s donor; φE0, which is related to the activity of the PSII reaction center’s acceptor; DIo/RC, which is related to the activity of the PSII reaction center; and PI*_ABS_*, a comprehensive characterization parameter that can characterize the photosynthetic performance of PSII (Figure 2). In particular, PSII potential activity (Fv/Fo), the quantum yield of absorbed light energy for electron transfer in PSII reaction centers (φE0), the PSII performance index based on absorbed light energy (PI*_ABS_*), and the heat dissipation per reaction center (DIo/RC) of PSII were all reduced by UV-B stress.

### 2.3. Examining the Enrichment Routes for MYB Transcription Factors in R. chrysanthum during UV-B Stress

Using the DNBSEQ platform, the transcriptomics assay measured 76.65 Gb of data in total. After removal of redundancy and assembly, 47,598 unigenes were generated. The unigenes were then aligned to seven functional databases for annotation; Table 1 displays the specifics of each sample. With 1328 coding transcription factors, unigenes were predicted using TransDecoder. A total of 46,940 genes were found using quantitative gene analysis, and 1915 of those genes had significant changes as a result of UV-B stress.

The *R. chrysanthum* was found to have 28 classes of transcription factors responsive to UV-B stress based on transcriptomic assays. The five classes of transcription factors with the greatest number of transcription factors responsive to UV-B stress are depicted in Figure 3A, while the remaining classes are shown in the Appendix A. The study found that MYB transcription factors were the most enriched, with 154 transcription factors undergoing enrichment analysis. The outcomes demonstrated that the primary impact on phenylpropane production was caused by MYB transcription factors (Figure 3B). Two genes had upregulated and three downregulated expression, according to an expression study of five important genes enriched to the phenylpropane biosynthesis pathway (Figure 3C).

### 2.4. Lignin Production in R. chrysanthum under UV-B Exposure Is Regulated by MYB Transcription Factors

A total of 2148 metabolites were found using extensively focused metabolomics based on UPLC–MS/MS assay technology and a self-constructed database of Jingjie PTM BioLab (Hangzhou, China) Co., Inc. Among them, 706 metabolites showed significant changes under UV-B stress.

We investigated further how MYB transcription factors influence lignin biosynthesis to enable *R. chrysanthum* to endure UV-B exposure. Based on the phenylpropanoid biosynthesis shown in Figure 3, lignin biosynthesis is one of the downstream metabolic processes of phenylpropanoid biosynthesis. To better illustrate the changes in associated metabolites and gene expression along the system, the lignin biosynthesis pathway was plotted (Figure 4). The results showed that four genes of the 4CL family (TRINITY_DN1438_c0_g2_i1-A1, TRINITY_DN24905_c0_g1_i1-A1, TRINITY_DN4521_c0_g2_i1-A1, and TRINITY_DN4521_c1_g1_i1-A1) were upregulated, and of the lignin biosynthesis-specific genes, five genes of the CCR family (TRINITY_DN21758_c0_g1_i1-A1, TRINITY_DN4801_c0_g1_i1-A1, TRINITY_DN4801_c0_g1_i2-A1, TRINITY_DN4801_c0_g1_i3-A1, TRINITY_DN955_c0_g2_i2-A1), two genes of the CAD family (TRINITY_DN16398_c0_g1_i1-A1, TRINITY_DN4452_c0_g1_i3-A1), and seven genes of the E1.11.1.7 family (TRINITY_DN19225_c0_g1_i1-A1, TRINITY_DN2204_c0_g4_i1-A1, TRINITY_DN2579_c1_g1_i3-A1, TRINITY_DN7904_c0_g1_i1-A1, TRINITY_DN7904_c0_g2_i1-A1, TRINITY_DN8137_c0_g1_i2-A1, TRINITY_DN9395_c0_g1_i4-A1) were upregulated. Detailed information on the four family genes is provided in Appendix A; transcription factors were named with reference to NR annotations and the Arabidopsis database. One MYB transcription factor (*RcTRP5*) was downregulated. The above results suggest that the negative regulatory effect of RcTRP5 on the lignin biosynthesis pathway is caused by the alteration of one of the differential genes or the common alteration of several genes in these pathways. We observed that the changes in the expression of the RcTRP5 transcription factor were opposite to those of the key genes in the pathway. To further explain the negative regulatory effect of RcTRP5 on the lignin biosynthesis pathway, the key enzymes involved in the pathway were analyzed for acetylation modification.

### 2.5. Changes in the Acetylation of Associated Enzymes in the R. chrysanthum Lignin Production Pathway during UV-B Stress

The acetylation proteomics of *R. chrysanthum* identified 9371 acetylation sites on 3754 proteins, of which 5063 sites on 1991 proteins contained quantitative information. We then conducted a systematic bioinformatics analysis of proteins with quantitatively informative sites. The five important enzymes shown in Figure 4’s lignin biosynthesis pathway had their acetylation modifications analyzed. The majority of the important enzymes in the lignin biosynthesis pathway, such as CAD1 (183), E1.11.1.7 (68), CCR1 (165), CAD (43), CCR1 (147), 4CLL7 (566), E1.11.1.7 (275), 4CLL7 (328), CCR1 (40), and PAL (391), were shown to have acetylation upregulation, according to the data (Figure 5A). Among them, there was a considerable upregulation of PAL (391), CAD (43), and E1.11.1.7 (68) (Figure 5B–D).

### 2.6. Building Three-Dimensional Structures of Essential Enzymes in the R. chrysanthum Lignin Production Pathway and Examining Their Non-Covalent Interactions

Our three-dimensional structures of PAL, CAD, and E1.11.1.7 were built using the three enzymes that were screened above and showed notable alterations in acetylation modifications. We also labeled the acetylation site (Figure 6). We examined the three enzymes’ salt-bridge structures and water-transport clusters in order to gain a molecular understanding of the characterization and function of the proteins. According to the findings, PAL has 13 hydrophobic clusters, and the areas of these clusters range from 76.0 to 5638.7 Å (Figure 6A). There were two to forty residues and two to one hundred and eighty-eight interactions among the residues. Based on calculations, 74 salt bridges with a K (kappa value) of 0.14 and an FCR (fraction of charged residues) of 0.29 were found in PAL. The hydrophobic cluster areas of the eight hydrophobic clusters in CAD range from 79.6 to 2887.0 Å (Figure 6A). There were two to twenty-one residues and two to sixty-two interactions among the residues. Based on calculations, it was found that CAD had 12 salt bridges with a K (kappa value) of 0.16 and an FCR (fraction of charged residues) of 0.22. Eight hydrophobic clusters with hydrophobic cluster sizes ranging from 58.3 to 2944.1 Å are seen in E1.11.1.7 (Figure 6C). There were between two and twenty residues total, and there were between two and sixty-four interactions among the residues. Nine salt bridges with a K (kappa value) of 0.18 and an FCR (fraction of charged residues) of 0.12 can be found in E1.11.1.7, according to calculations.

### 2.7. Analyzing R. chrysanthum Physiological and Bioinformatics Characteristics in Connection with UV-B Stress

The expression of the enzyme genes, particularly those corresponding to *RcTRP5*, exhibited a significant negative correlation with the acetylation modification of the key enzyme for lignin synthesis (Figure 7A). These correlations were 0.91, 0.92, and 0.88 with PAL, CAD, and E1.11.1.7, respectively, indicating that the RcTRP5 transcription factors modulate the acetylation modification of the enzyme and thereby affect lignin biosynthesis.

Based on the findings of the correlation between physiological parameters and the acetylation modifications of key lignin synthesis enzymes, CAD and PAL were found to be positively correlated with Fv/Fo and DIo/RC, respectively. This suggests that the acetylation modifications of these two enzymes are connected to the possible activity of PSII (Figure 7B).

The results showed that the acetylation modification of E1.11.1.7 was primarily positively connected with the expression of G-type lignin, with higher monomeric G/S, based on the correlation between acetylation modifications of key enzymes of lignin production and lignin concentration (Figure 7C).

### 2.8. Building a Model Map to Illustrate How R. chrysanthum Reacts to UV-B Stress

Overall, UV-B stress resulted in the simultaneous destruction of both the donor and acceptor sides of PSII in *R. chrysanthum*, which led to the upregulation of acetylation modification of the K68 locus of the *RcTRP5* transcription factor in *R. chrysanthum*, and the downregulation of the expression level of the *RcTRP5* transcription factor indirectly promoting the acetylation modification of the CAD and PAL enzymes, with the upregulation of the K43 locus of CAD and the downregulation of the K83 locus of PAL, leading to the elevation of G-type lignin expression site of the CAD enzyme and downregulation of the K391 site of the PAL enzyme, which in turn led to elevated G-type lignin expression and elevated G/S (Figure 8). The reduced damage brought on by UV-B stress in *R. chrysanthum* is further demonstrated by the increased expression of lignin, a material that absorbs UV light.

## 3. Discussion

Rich physiological information about the plant can be found in the OJIP curve. The various stages of the curve depend on the redox state of PSII, which can reflect various aspects of the information such as the electron transfer process, the primary photochemical reaction of PSII, the electron transfer state of the photosynthetic apparatus, the activity of the PSII donor and acceptor, changes in the environment, the allocation of bioenergy, the size of the PQ pool of the electron transfer, and the activity of photosynthesis on the leaves [20]. The mechanism of action of UV-B stress on the alterations in the electron transport chain and the photosynthetic efficiency of *R. chrysanthum* were examined in this work using the rapid chlorophyll fluorescence technique. The OJIP curve under UV-B stress was flatter than that of the control group (Figure 1A) due to the increase in Fo (O phase) and decrease in Fm (P phase). The O point’s fluorescence magnitude was correlated with both the amount of antenna pigment content and the reaction center’s active state, which was inactivated in the PSII reaction center, while Fo rose under UV-B stress. Phase I was linked to the inactivation of FNR, a terminal oxidase of the photosynthetic electron chain that reduces NADP^+^ to NADPH by accepting electrons. The increased fluorescence intensity of phase I was suggestive of FNR’s inactivation (Figure 8), which is consistent with the proteomics results. The PSII reaction center is fully closed, all primary electron acceptors of PSII and Q_A_ are completely reduced, and the electron flow on the PSII acceptor side is saturated. The excessive energy used in the electron transfer to the slow-reducing PQ pool is the cause of the P-phase’s decrease in fluorescence intensity. In conclusion, the PSII reaction centers of *R. chrysanthum* showed inhibition of electron transfer on the donor side (Figure 2A) and acceptor side (Figure 2B) due to UV-B stress, and an increase in QA^−^ accumulation resulted in a reduction in the relative electron transfer rate. Consistent with earlier findings in cucumber under salt stress [21], the rise in DIo/RC reflects an increase in the proportion of total dissipation of active RCs because of increased dissipation of inactive RCs [22]. Due to damage to *R. chrysanthum*’s photosynthetic machinery, the efficiency of light energy conversion is reduced, as seen by the drop in PI*_ABS_* under UV-B. In comparison to the other photosynthetic metrics, the change in PI*_ABS_* is more sensitive, representative, and observable [20].

Plants have a vast family of transcription factors, including MYB transcription factors. The increasing body of research suggests that plant MYB transcription factors play critical roles in secondary metabolism, physiological regulation, growth and development, and other functions. The ability of MYB transcription factors to withstand abiotic stress is crucial [23]. In this study, *RcTRP5* belongs to the MYB transcription factor family. In response to UV-B stress, the expression level of *RcTRP5* is reduced, which negatively regulates the phenylpropane biosynthesis of *R. chrysanthum* (Figure 4). The results were similar to those of alfalfa *MYB17* transcription factor negatively regulating lignin and flavonoid synthesis [24].

Transcription factors, including as NAC, MYB, WRKY, and other family transcription factors, control the production of lignin. The MYB family genes are the lignin synthesis transcription factors that have been investigated the most. The lignin synthesis pathway is transcriptionally repressed by the maize *ZmMYB31* and *ZmMYB42* transcription factors, as demonstrated by earlier research. Transgenic *Arabidopsis thaliana* that overexpressed *ZmMYB42* exhibited downregulated expression of the F5H, C4H, and 4CL genes, decreased lignin content, and increased G/S [25].To explore the molecular role of *RcTRP5* and its mechanism, we subjected UV-B-stressed *R. chrysanthum* to transcriptomics and metabolomics sequencing investigation. Enrichment analysis of MYB transcription factors showed that they were mainly enriched in the phenylpropane biosynthesis pathway of *R. chrysanthum* (Figure 3B), which was indicated to negatively regulate phenylpropane biosynthesis due to the decreased expression level of *RcTRP5* transcription factor. Metabolomics showed that the expression of G-type lignans was elevated, the expression of S-type lignans was decreased, and G/S was elevated, which was consistent with the level of PAL acetylation modification, and further verified that the *RcTRP5* transcription factor negatively regulated lignan biosynthesis (Figure 4).

Numerous biological processes have been shown to involve the function of Kac changes, most notably histone acetylation [26]. Though the amount of acetylation in plant proteins has been revealed by recent research in model organisms, little is known about the impact of acetylation modifications that do not occur on histones. We discovered many acetylation sites in proteins involved in lignin biosynthesis under UV-B exposure, which is similar to results in other plants and suggests the significance of Kac modification for cellular function. Furthermore, we observed that some MYB transcription factors are lysine-acetylated proteins (Figure 5D), and that controlling the activity of the *RcTRP5* transcription factor in *R. chrysanthum* against UV-B stress may be aided by Kac modification. In addition, salt bridges and hydrophobic structures play an important role in maintaining the three-dimensional structure of proteins (Figure 6). Because the function of a protein depends on its three-dimensional structure, after determining the protein structure of the target, its salt bridge and hydrophobic structures were further characterized to better respond to how these proteins function under UV-B stress. Ultimately, we found that acetylation of the K68 site of the *RcTRP5* transcription factor is upregulated, which in turn negatively regulates lignin biosynthesis.

The results of this study are of great significance for the conservation of *R. chrysanthum* germplasm resources and provide a new perspective for understanding the changing patterns of metabolites and their regulatory mechanisms in plants under adverse conditions. In addition, a deeper understanding of the adaptation mechanisms of plants to UV-B radiation is crucial for the maintenance of biodiversity, especially in ecosystems that are extremely sensitive to changes in UV-B radiation. As global climate change continues, these findings also have long-term implications for enhancing agricultural productivity.

## 4. Materials and Methods

### 4.1. Plant Material, Growing Conditions, and Treatments

The Changbai Mountain (40.10° N, 100.10° E) *R. chrysanthum* was the subject of the experiment. It was cultivated in 1/4 MS solid medium and kept in a smart artificial climate chamber that was set to 18 °C/16 °C (day/night), with a 14 h/10 h photoperiod (day/night) and 60% relative humidity. Six (*n* = 3) genetically identical *R. chrysanthum* seedlings with comparable morphologies were chosen, and three of them were split into two groups to serve as a set of biological duplicates.

The radiation treatments we applied to the seedlings were a slight modification of earlier radiation treatments we performed in our laboratory using PAR and PAR + UV-B light [27]. To administer the PAR treatment, a 400 nm long-pass filter (Edmund, Filter Long 2IN SQ, Barrington, NJ, USA) was placed above the culture bottle, ensuring that the *R. chrysanthum* would only receive PAR light. For the PAR + UV-B treatment, a 295 nm long-pass filter (Edmund, Filter Long 2IN SQ, Barrington, NJ, USA) was placed above the culture vials. The radiation treatment lasts for a full 48 h. During the radiation process, a synthetic UV-B lamp (Philips, Ultraviolet-B TL 20 W/01 RS, Amsterdam, the Netherlands) was employed. The samples received effective irradiances of 2.3 W·m^−2^ for UV-B and 50 μmol·m^−2^·s^−1^ for PAR, respectively, depending on the transmission function of the long-pass filter. The application of a UV intensity meter (Sentry Optronics Corp., ST-513, SHH, New Taipei City, China) and an illuminance meter (TES Electrical Electronic Corp., Tes-1339 Light Meter Pro., Taipei, China) for monitoring.

### 4.2. Measurement of OJIP Transients and Analysis of Fast Fluorescence-Induced Kinetics

On both completely grown and still attached leaves, chlorophyll (Chl) fluorescence transient (OJIP) profiles were measured. With a Handy-PEA (Hansatech Instruments Ltd., King’s Lynn, UK) for monitoring and analysis, leaf clamps were applied to the leaf at an average of three detection points. Following a half-hour dark adaptation period, the instrument probe was fastened to the leaf clamps, and the slide switch was activated to expose the measurement holes to the laser light source. For quick acquisition of fluorescence signals, an LED light source with a 3000 μmol·m^−2^·s^−1^ preset detection time of 1 s was employed.

When every PSII reaction center (RC) was activated, the minimum fluorescence intensity signal (Fo) was measured at 20 μs (O phase), and when every PSII RC came closer to the maximum intensity signal (Fm), it was measured at 200–500 ms (P phase). F_J_ and F_I_ represent the fluorescence intensity at 2 ms (J phase) and 30 ms (I phase).

Minimum fluorescence intensity: (Fo);

Potential photochemical efficiency: [Fv/Fo = (Fm − Fo)/Fo];

Electron transfer absorbed energy quantum yield: [φ_E0_ = [1 − (Fo/Fm)](1 − Vj)];

Heat dissipation per unit reaction center: [DIo/RC = ABS/RC − TRo/RC];

Performance parameters based on absorbed light energy: [PI*_ABS_* = (RC/ABS) $\frac{\varphi P0}{1-\varphi P0}\frac{\psi 0}{1-\psi 0}$]

### 4.3. UPLC–MS/MS-Based Determination of Metabolites of R. chrysanthum

Metware Biotechnology Co. (Wuhan, China) was used to carry out the metabolic analysis in accordance with the following protocol: Using vacuum freeze-drying technique, the cowhide azalea samples were placed in a lyophilizer (Scientz-100F) and then ground into powder form (30 Hz, 1.5 min) using a grinder (MM 400, Retsch, Haan, Germany). Then, 50 mg of sample powder was weighed using an electronic balance (MS105DΜ), 1200 μL of −20 °C pre-cooled 70% methanol aqueous solution of internal standard extract (Merck Company, Darmstadt, Germany) was added, and the supernatant was removed and passed through a 0.22 μm membrane filter. Tandem mass spectrometry (https://sciex.com.cn/) and a UPLC–ESI–MS/MS system (UPLC, ExionLC™ AD, https://sciex.com.cn/) were used to analyze the sample extracts. Analyses of all sample extracts were conducted using the previously mentioned standard protocols [28].

Identification was based on secondary spectra of fragment ions from the Metware database. The relative content of metabolites was computed using peak regions. By using the orthogonal partial least squares discriminant analysis (OPLS-DA) method, metabolites with VIP > 1 were classified as differential, those with FC > 1.2 as significantly upregulated, and those with FC < 0.67 as significantly downregulated. The KEGG database was used to enrich and classify metabolites.

### 4.4. Transcriptomic Assay of R. chrysanthum

BGI Genomics Co., Ltd. (Shenzhen, China) carried out the transcriptomics assay for *R. chrysanthum* in this study. The precise experimental analysis protocol involved processing total RNA via mRNA enrichment and purifying mRNA with poly(A) using Oligo(dT) magnetic beads and Oligotex mRNA Kit(Hangzhou Liankemeixun Biomedical Technology Co. (Hangzhou, China)). Lysis is performed at elevated temperatures while a suitable quantity of lysing agent is added. The first strand of cDNA is created using the interrupted mRNA as a template, and the second strand is created using the configure synthesis reaction system. The sticky ends are recovered, cleaned, and repaired using the proper supplies. The purified cDNA’s mucilaginous end is joined to the “A” at the “3” end. After choosing the changed products based on fragment size, polymerase chain reaction (PCR) amplification was carried out. An Agilent 2100 Bioanalyzer (Shanghai Asiagene Technology Co., Ltd (Shanghai, China)) and ABI StepOnePlus Real-Time PCR equipment(Hangzhou Liankemeixun Biomedical Technology Co. (Hangzhou, China)) were used to evaluate the quality of the created libraries.

Reads with more than 10% unknown base N content, contamination, and low-quality reads (reads with a quality value of less than 15 bases accounting for more than 50% of the total number of bases in that read are considered low-quality reads) were filtered and removed using SOAPnuke to ensure data quality and accuracy. The Bowtie2 program was used to align each sample’s gene expression levels to the reference gene sequences. The RESM (v1.2.8) software package was then utilized to compute the expression levels of each sample.

To detect DEGs in *R. chrysanthum* in response to UV-B stress, the DEseq R program (http://www.bioconductor.org/packages/release/bioc/html/DESeq2.html (accessed on 9 June 2024)) was employed. Qvalue (adjusted *p* value) < 0.05 was regarded as a DEG in this investigation. The analysis of the previously specified standard processes was conducted strictly in accordance with this experimental protocol [29].

### 4.5. Proteomic Analysis of Acetylation-Modified Proteins in R. chrysanthum

Jingjie PTM BioLab (Hangzhou, China) Co., Inc. conducted the proteome and acetylation modification proteomics in this work. Using an organic combination of many state-of-the-art methods, including mass-spectrometry-based quantitative proteomics and off-label quantification, the samples’ quantitative proteomic analysis was completed. The usual protocols previously mentioned were adhered to during the experimental procedure [30].

The specific experimental procedure for acetylation modification proteomics is as follows:

Selected *R. chrysanthum* samples were pulverized to a fine powder in a mortar that had been chilled in liquid nitrogen beforehand. After adding four times the powder’s volume to each tube containing the sample in phenol extraction buffer (10 mM dithiothreitol and 1% protease inhibitor), the sample was ultrasonically lysed. After centrifuging the phenol extraction solution at 5500× *g* for 10 min at 4 °C, an equal volume of Tris solution was added to equilibrate it. After taking the supernatant, five times its volume of 0.1 M ammonium acetate/methanol was added, allowed to precipitate for an entire night, and then the mixture was washed with acetone and methanol, respectively. Using the BCA kit (Beyotime Biotechnology (Shanghai, China)), the precipitated protein was re-solubilized with 8 M urea, and the protein concentration was ascertained.

Dithiothreitol was added to the protein solution that had been treated in the previous step to bring the mixture’s final concentration to 5 mM. The combination was then treated for 30 min at 56 °C to decrease the protein. To make the concentration 11 mM, iodoacetamide was added to the mixture. It was then incubated for 15 min at room temperature under light protection. Afterwards, trypsin (trypsin: protein = 1:50 by mass) was added and the mixture’s urea content was lowered to less than 2 M. The digestion process was then left to occur overnight at 37 °C. Trypsin was reintroduced after the treatment at a ratio of trypsin to protein of 1:50, and the enzymatic digestion was allowed to continue for an additional four hours.

The peptides that had undergone trypsinization were desalted using Strata X C18 (Phenomenex, Torrance, CA, USA), freeze-dried under vacuum, dissolved in 0.5 M TEAB, and labeled in compliance with the guidelines provided by the TMT kit (Thermo Fisher Scientific (Shanghai, China)).

The peptides were dissolved in liquid chromatography mobile phase A, separated using an EASY-nLC 1000 ultra-high performance liquid chromatography system (Thermo Fisher Scientific (Shanghai, China)), and injected into an NSI ion source for ionization at a voltage of 2.0 kV. After ionization, the peptides were analyzed by Q Exactive mass spectrometry. Data acquisition was performed using a data-dependent scanning (DDA) program(Beijing Baitai Paike Biotechnology Co., (Beijing, China)). In order to optimize the efficient use of the mass spectra, the signal threshold was set to 20,000 ions/s, the maximum injection time was set to 100 ms, the automatic gain control (AGC) was set to 1 × 10^5^, and the dynamic exclusion time for tandem mass spectrometry scans was set to 30 s in order to prevent repeated scans of the parent ion. Ultimately, proteins were deemed to be considerably upregulated if they had *p* < 0.05 and FC > 1.5.

### 4.6. Data Analysis

Each of the aforementioned studies had three biological replications and a fully randomized design. Utilizing IBM SPSS Statistics 26 (IBM Corporation, New York, NY, USA) software, the fluorescence and histological properties of *R. chrysanthum* were examined using Duncan’s method of one-way difference between samples for significance (*p* < 0.05).

## 5. Conclusions

In this study, we used physiological data and combined transcriptomics, metabolomics, and acetylation-modified proteomics analyses to search for a *RcTRP5* transcription factor responsive to UV-B stress. We also visualized the lignin biosynthesis pathway, discovered that acetylation was upregulated at the *RcTRP5* transcription factor’s K68 site, and concluded that the *RcTRP5* transcription factor negatively regulates lignin biosynthesis. Furthermore, acetylation modification of two essential lignin biosynthesis-related enzymes, PAL and CAD, was discovered. Based on these findings, we postulated that the indirect promotion of acetylation modification of the two enzymes is caused by downregulation of *RcTRP5* transcription factor expression, which in turn influences lignin biosynthesis. This work offers important new understandings of how MYB transcription factors control the manufacture of lignin in *R. chrysanthum* under UV-B stress. It also offers theoretical justification for increased plant quality and UV resistance.

## Figures and Tables

**Figure 1 ijms-25-09205-f001:**
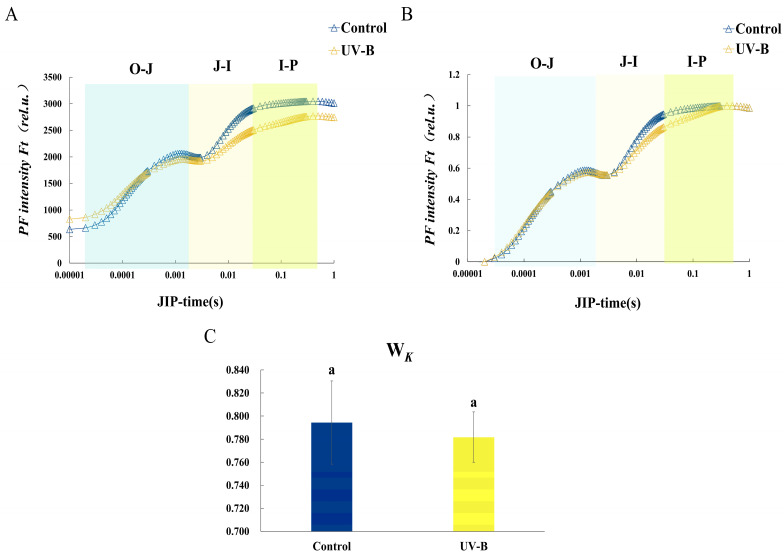
Chlorophyll fluorescence (OJIP) transient curves and standardized curves illustrating *R. chrysanthum’s* response to UV-B stress. (**A**). Modifications in the OJIP transient curves following UV-B stress treatment in *R. chrysanthum*. (**B**). OJIP normalized curves with UV-B stress control applied. The normalization of O–P segments was based on W_O−P_ = (F_t_ − F_O_)/(F_m_ − F_O_). (**C**). The ratio of the relative variable fluorescence value to the amplitude of F_J_−F_O_ is represented by the formula W_K_ = (F_K_ − F_O_)/(F_J_ − F_O_). Means ± SD are represented by the values (*n* = 3). Letter (a) indicate a significant difference at *p* < 0.05 among treatments.

**Figure 2 ijms-25-09205-f002:**
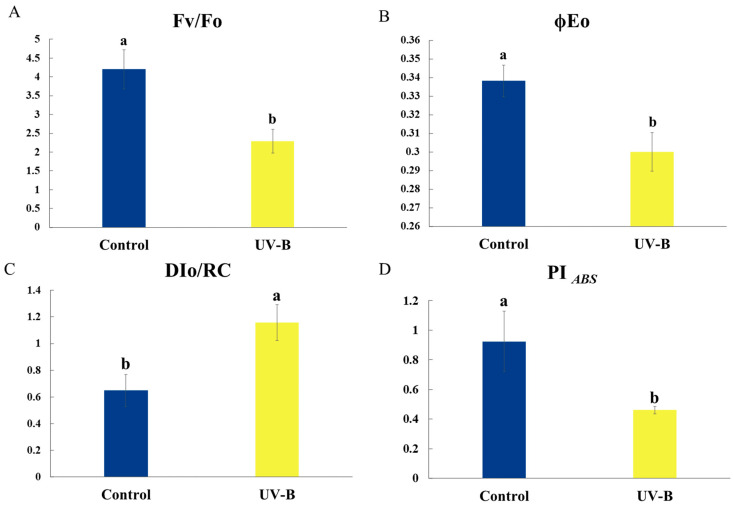
JIP parameter variations in *R. chrysanthum* in response to UV-B stress. (**A**–**D**). According to *R. chrysanthum*’s reaction to UV-B stress, it shows changes in the PSII donor side, PSII reaction center, PSII acceptor side, and PSII light and performance parameters. Values are means ± SD (*n* = 3). Different letters (a, b) indicate a significant difference at *p* < 0.05 among treatments.

**Figure 3 ijms-25-09205-f003:**
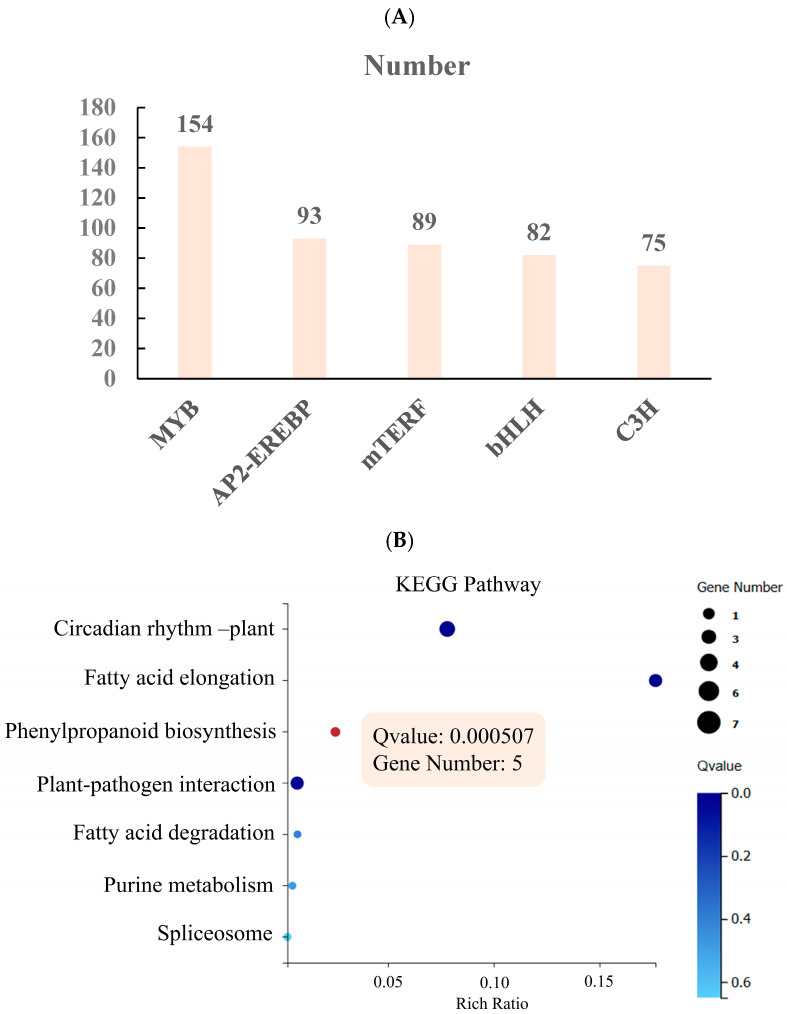
Classification of transcription factors and analysis of MYB transcription factor enrichment in *R. chrysanthum* under UV-B stress. (**A**) Categorization of *R. chrysanthum* transcription factors in reaction to UV-B stress. (**B**) Analysis of MYB transcription factors in *R. chrysanthum* through enrichment. The size of the solid dots indicates the number of genes enriched into the pathway, and the color of the solid dots indicates the significance of the pathway; the closer the color is to dark blue, the more significant it is. (**C**) Key genes in the phenylpropane biosynthesis pathway and their corresponding changes in expression; red indicates upregulation and green indicates downregulation, the control group is shown by the inner diameter, and the UV-B-stress-treated group is shown by the outside diameter.

**Figure 4 ijms-25-09205-f004:**
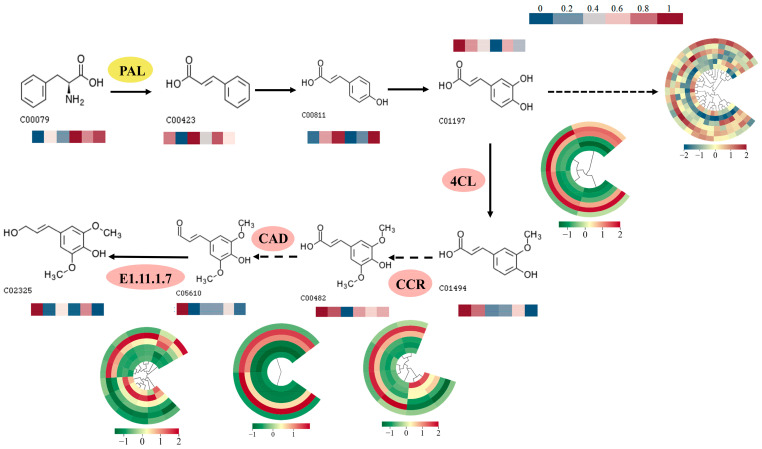
MYB transcription factors control the biosynthesis of lignin in *R. chrysanthum* under UV-B stress. PAL: phenylalanine ammonia-lyase; 4CL: 4-coumarate:CoA ligase; CCR: cinnamoyl-CoA reductase; CAD: cinnamyl alcohol dehydrogenase; E1.11.1.7: peroxidase. Data on metabolite content and gene expression were plotted on a heat map after being normalized using the formula (X*i* − min(*x*))/(max(*x*) − min(*x*)). A bar heat map is used to show changes in metabolite content, with bluer colors denoting lower levels and redder colors representing higher levels. Changes in gene expression are represented by a circular heat map, in which the three biological replicates of the control and the three biological replicates of the UV-B stress treatment are indicated sequentially from the inner to the outer layers, with redder colors indicating higher expression and greener colors indicating lower expression.

**Figure 5 ijms-25-09205-f005:**
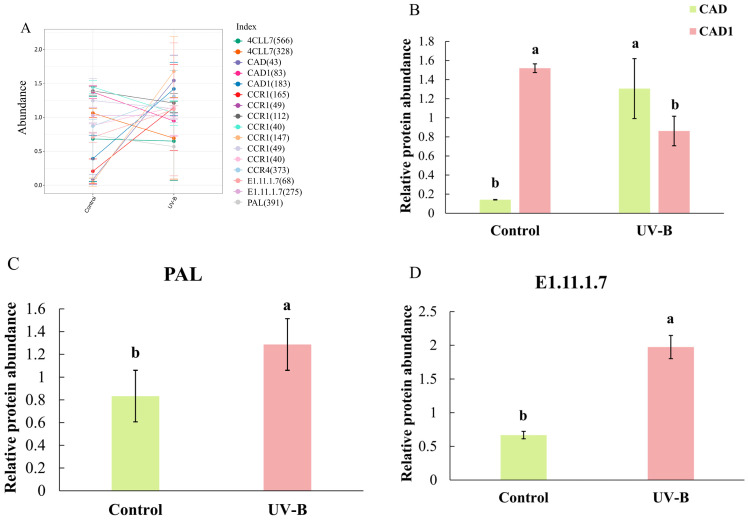
Under UV-B exposure, the *RcTRP5* transcription factor of *R. chrysanthum* modifies the acetylation of related enzymes in the lignin production pathway. (**A**) The lignin biosynthesis pathway’s enzymes are impacted by acetylation variations. The locations of protein acetylation are shown in parentheses. (**B**–**D**) Key enzymes in the lignin biosynthesis pathway are altered by acetylation changes. GSVIVT00023967001: peroxidase. Values are means ± SD (*n*= 3). Different letters (a, b) indicate a significant difference at *p* < 0.05 among treatments.

**Figure 6 ijms-25-09205-f006:**
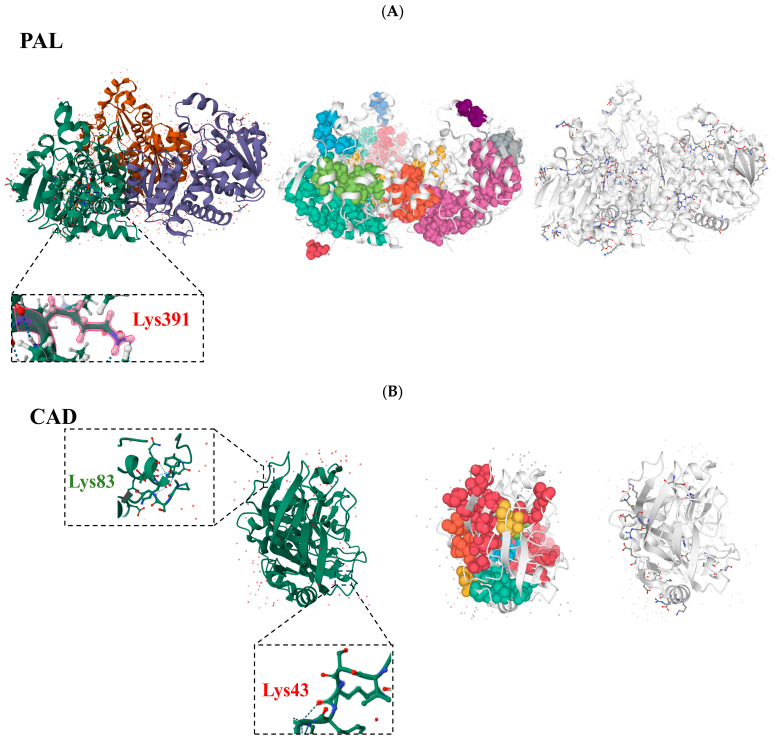
Key acetylation modifications in the lignin biosynthesis pathway, which is controlled by the *RcTRP5* transcription factor in *R. chrysanthum* under UV-B stress, and their three-dimensional structure. (**A**) Acetylation site labeling, hydrophobic clusters, and salt bridges for phenylalanine ammonia-lyase, in that order from left to right. (**B**) Acetylation site labeling, hydrophobic clusters, and salt bridges for cinnamyl alcohol dehydrogenase, in that order from left to right. (**C**) Hydrophobic clusters, salt bridges, and peroxidase acetylation site labeling are arranged from left to right.

**Figure 7 ijms-25-09205-f007:**
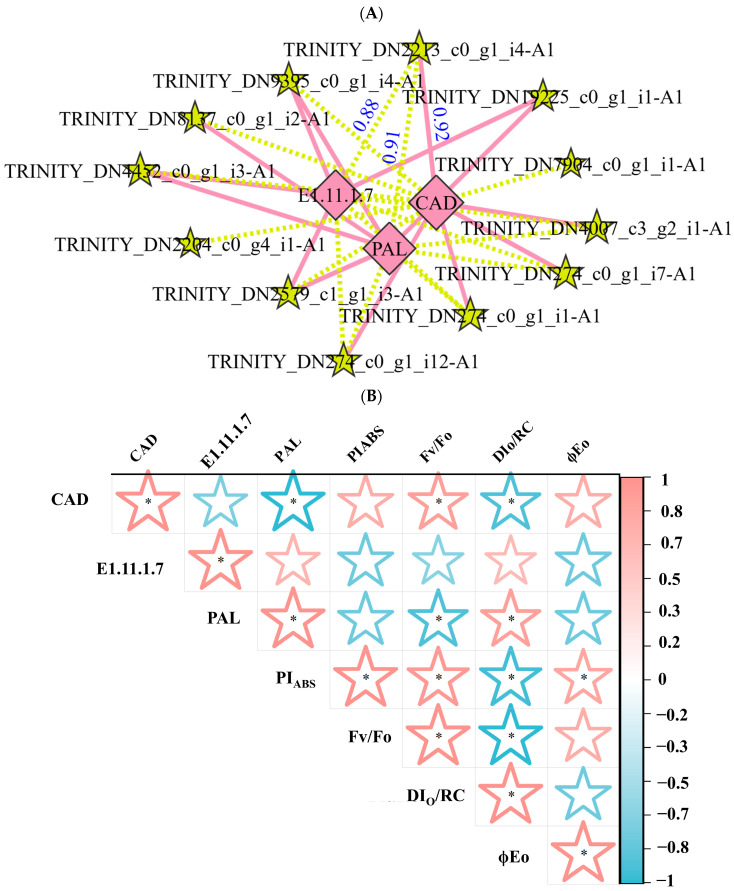
Correlation study of *R. chrysanthum* physiological and histological characteristics under UV-B stress. (**A**) An examination of the relationship between the expression of lignin synthase genes and the acetylation modification of important lignin synthesis enzymes in *R. chrysanthum* under UV-B stress. Pentagrams and diamond squares represent metabolites and genes, respectively; solid pink lines indicate positive correlations and dashed yellow lines indicate negative correlations; numbers indicate correlation coefficients. (**B**) The expression of acetylation modifications of important lignin synthesis enzymes and photosynthesis parameters in *R. chrysanthum* under UV-B stress were correlated. (**C**) Examining the relationship between *R. chrysanthum* under UV-B stress and the expression of acetylation modifications of important lignin synthesis enzymes. Values are means ± SD (*n* = 3). The asterisk (*) indicate a significant difference at *p* < 0.05 among treatments.

**Figure 8 ijms-25-09205-f008:**
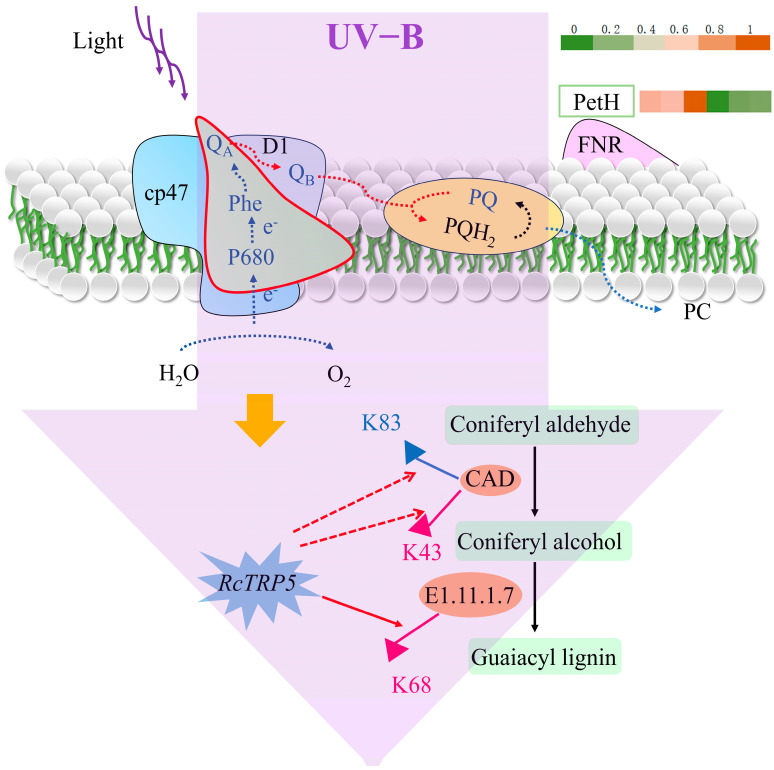
The response of *R. chrysanthum* to UV-B stress in an illustration. The red solid line in PSII indicates impaired light and performance after UV-B radiation, the red dashed line in the pathway indicates indirect upregulation, the red solid line indicates direct upregulation, the pink arrowheads indicate upregulation of the acetylation site, and the blue arrowheads indicate downregulation of the acetylation site.

**Table 1 ijms-25-09205-t001:** Detailed information on transcriptomics testing for each sample.

Sample	Total Raw Reads (M)	Total Clean Reads (M)	Total Clean Bases (Gb)	Clean Reads Ratio (%)
Control 1	43.69	42.55	6.38	97.38
Control 2	43.69	42.34	6.35	96.9
Control 3	43.69	42.39	6.36	97.03
UV-B 1	43.69	42.39	6.36	97.03
UV-B 2	43.69	42.37	6.36	96.97
UV-B 3	45.44	43.46	6.52	95.64

## Data Availability

The data used in this study are available from the corresponding author on submission of a reasonable request.

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
