# Peer review of "RcTRP5 Transcription Factor Mediates the Molecular Mechanism of Lignin Biosynthesis Regulation in R. chrysanthum against UV-B Stress"

_ijms, 2024, doi:10.3390/ijms25179205_

Round 1

Reviewer 1 Report

Comments and Suggestions for Authors

1.      L10-15 these sentences need to be condense and combine to point out the relationship between MYB-UVB-lignin.

2.      L30 “Since they cannot move, plants must photosynthesize in order to maintain life. ”. please re-write this sentence

3.  Figure3B, what does the gene number mean (black circles)? Figure3C,4A these figures are hard to understand, including figure legend. What are the y-axis of these figures?

4.  Figure8, what does these lines (red,pink,dotted..) mean in the figure?

5.  L312-321 many of these should be put in the introduction.and most of them seem to have no relationship with this study?

6.  RcTRP5 is the only gene that was decreased by UV treatment?

7.  The discussion need to be largely improved, since it fail to decipher the results obtained of this study.

Comments on the Quality of English Language

need to be improved

Reviewer 2 Report

Comments and Suggestions for Authors

The manuscript investigates the response of Rhododendron chrysanthum to UV-B stress, focusing on its impact on photosynthesis and lignin biosynthesis. The study combines physiological, biochemical, and molecular approaches to comprehensively understand how UV-B stress affects the plant's photosynthetic efficiency, MYB transcription factors, and lignin production.

1.     The introduction provides a solid background on the importance of photosynthesis and lignin biosynthesis in plants under UV-B stress. However, it would benefit from a more detailed discussion on the ecological and practical significance of R. chrysanthum, which would help contextualize the study.

2.     The roles of MYB transcription factors and lignin in stress responses are well-explained. The inclusion of specific examples, such as the role of AgMYB5 in celery and AtMYB102 in Arabidopsis, effectively highlights the relevance of these factors.

3.     The potential implications of the findings for improving plant resilience to UV-B stress are briefly mentioned. Expanding on these implications, particularly in the context of plant breeding and agricultural practices, would enhance the impact of the study.

Reviewer 3 Report

Comments and Suggestions for Authors

The study is original as it makes an integrative analysis combining transcriptomics and protein acetylation to find the regulation of UV-B stress. But in the present form the manuscript is difficult to follow, and the flow of results is rather obscure. The paper needs major improvement in the results sections. Some experiments are difficult to follow, or are missing essential controls.

For instance: RcTRP5 was considered to be the essential negative regulator because according to the genome wide data the behaviour is opposite to the myb pathway, But this should have been confirmed by RT PCR analysis to confirm the genome wide data.

Figure 5 is difficult to understand. What do the yellow and the pink bar mean? Please explain which experiment and results are described, increase of acetylation? What units are depicted in the y axis?

 Figure 1: There is no b in the graphic, so does not need to be defined in the legend. Is the UV treatment enough? differences are minimal and not significative.

Figure 2: R. chrysanthum in italics.

Line 30-32: This is a biological nonsense. The fact that plants photosynthetize is not because they are sessile, as long as there are flagellated or cilliated microorganisms able to photosynthetize. Euglena for instance. UV radiation is not harmful to plants because of the photosynthesis, is harmful to any living organism as is able to mutageneize DNA. Please correct.

Line 53: A. thaliana in italics.

Line 78: This is not the description of acetylation, as acetylation is the addition of an acetyl residue to any molecule. This is the description of protein acetylation. Please correct.

Round 2

Reviewer 3 Report

Comments and Suggestions for Authors

Papaer is now ready for acceptance. Congratulations